# An improved formula for the Complete Data Fusion

Simone Ceccherini, Nicola Zoppetti, and Bruno Carli

Istituto di Fisica Applicata "Nello Carrara" del Consiglio Nazionale delle Ricerche, Via Madonna del Piano 10, 50019 Sesto Fiorentino, Italy

*Correspondence to*: Simone Ceccherini (S.Ceccherini@ifac.cnr.it) and Nicola Zoppetti (N.Zoppetti@ifac.cnr.it)

**Abstract.** The Complete Data Fusion is a method that combines independent measurements of atmospheric vertical profiles. Recently a new formula for the Complete Data Fusion, which does not contain matrices that can be singular and overcomes the generalized inverse approximation used when singular matrices have to be inverted, has been proposed. We show that the new formula is a generalization of the original one and analyze the analytical relationship between the two formulas when

generalized inverse matrices are used for the inversion of singular matrices. We extend the new formula to include interpolation and coincidence errors, which must be considered when the profiles to be fused are measured on different vertical grids and at different times and/or locations. Finally, we use a real measurement of the IASI instrument to show the improved performances of the new formula with respect to the original one.

## 1 Introduction

The Complete Data Fusion (CDF) was firstly introduced in Ceccherini et al. (2015) as a new data processing method that allows to combine several independent measurements of an atmospheric vertical profile, and more generally of any vectorial quantity that is retrieved using the optimal estimation method (Rodgers, 2000). It is called "complete" for its capability of considering all the features of the measurements that are being combined, that is not only their errors, but their vertical resolution as well. The inputs of the method are the profiles retrieved from the individual measurements using the optimal

estimation method together with their a priori profiles, averaging kernel matrices (AKMs) and noise covariance matrices (CMs), and an a priori profile with its CM is used to constrain the fused profile. The output of the method is a single profile (the fused profile) with its AKM and CM. The a priori information used to constrain the fused profile can be freely chosen independently of the a priori information used in the retrievals of the individual profiles so that the method can also be used to change the a priori of a retrieved product. When in the variability range of the results of the individual retrievals the linear

approximation of the forward models is appropriate, the method is equivalent to the simultaneous retrieval of all the measurements that are combined (see Appendix of Ceccherini et al. (2015) for the proof). The implementation of the simultaneous retrieval requires the integration into a single inversion system of the different radiative transfer models that simulate the measurements of the different sensors and the access to the different (Level 1) measurements, implying the use of large computational resources specifically developed for each fusion operation. The CDF overcomes these complications

by combining the Level 2 products separately supplied by the different retrieval processors.

The method has been extended to fuse profiles retrieved on different vertical grids for which an interpolation on a common grid is needed and to deal with measurements obtained either at different times or from different platforms and, therefore, referred to different true profiles. This extension required the introduction of interpolation and coincidence errors in the fusion process (Ceccherini et al., 2018).

The performance of the method has been studied on ozone profiles retrieved from simulated measurements in the ultraviolet, visible, and thermal infrared spectral ranges for the Sentinel-4 and Sentinel-5 missions of the Copernicus program (Tirelli et al., 2020, Zoppetti et al., 2021). The results of these studies show that the CDF is able to provide products of improved quality with respect to the input products in terms of reduced errors and increased number of degrees of freedom.

A problem connected with the application of the CDF formula is the presence of the inverse matrices of the noise CMs of the
input profiles and this implies that the formula can be rigorously applied only when the noise CMs are nonsingular. When
the profiles are retrieved solving ill-posed inverse problems (which is a very common case), this condition is not satisfied. In
this case, we can still apply the CDF formula replacing the inverse matrices of the noise CMs with the generalized inverse
matrices (Kalman, 1976), but the result is an approximation. Furthermore, a practical problem in the use of the generalized
inverse matrices is the definition of the threshold for the eigenvalues for which eigenvalues smaller than this threshold have
their inverses replaced with zeros. Too small values for this threshold determine significant numeric noise in the products; on
the other hand, too large values of this threshold determine a loss of useful information.

Recently, following the approach of the Kalman filter (Kalman, 1960, Rodgers, 2000) as done in Schneider et al. (2022), a
different formula for the CDF has been derived (Ceccherini, 2022) for the fusion of two profiles. This formula contains the
inverse matrices of the retrieval error CMs, which include both the noise and the smoothing errors, instead of the inverse
matrices of the noise CMs. Differently from the noise CMs, the retrieval error CMs are always nonsingular matrices and the
new formula can be used without having to resort to the use of generalized inverse matrices.

In this paper we extend the new formula to the fusion of any number of profiles and show that it is a generalization of the
original CDF formula given in Ceccherini et al. (2015). Furthermore, we analytically analyze the differences between the
new formula and the original one when the generalized inverse matrices are used for the inverse of the noise CMs. Since in
the application of the CDF to real measurements it is common practice to interpolate between different grids and to consider
not perfect coincidence of the fusing profiles, the new formula is also used to derive the operational expression that takes
into account interpolation and coincidence errors.

Finally, we use a measurement of the IASI instrument (Clerbaux et al., 2009) to show the improved performances of the new
formula with respect to the original one in the case of real data.

In Section 2, we show that the new formula is a generalization of the original one and extend it to handle the cases where
coincidence and interpolation errors are present. In Section 3, we compare the performances of the two formulas using an
IASI measurement and in Section 4 we draw the conclusions.

## 2 Theoretical analysis of the CDF formula

### 2.1 The new formula as a generalization of the original one

We assume to have $N$ profiles $\hat{\mathbf{x}}_i$ retrieved on the same vertical grid with the optimal estimation method (Rodgers, 2000)

from $N$ independent measurements of a true atmospheric profile $\mathbf{x}_t$. The profiles $\hat{\mathbf{x}}_i$ are characterized by the AKMs

$\mathbf{A}_i = \dfrac{\partial \hat{\mathbf{x}}_i}{\partial \mathbf{x}_t}$, which determine the sensitivities of the profiles $\hat{\mathbf{x}}_i$ to $\mathbf{x}_t$ and by the CMs $\mathbf{S}_i$, which determine the retrieval

errors.

Before introducing the new formula for the CDF, let us recall some useful relationships. The quantities $\mathbf{A}_i$ and $\mathbf{S}_i$ can be
written as a function of the two quantities that characterize the retrievals, that is the Fisher information matrices (Fisher,
1935) $\mathbf{F}_i = \mathbf{K}_i^T \mathbf{S}_{\mathrm{ny}_i}^{-1} \mathbf{K}_i$ ( $\mathbf{K}_i$ being the Jacobian matrices of the forward models and $\mathbf{S}_{\mathrm{ny}_i}$ the CMs of the noise errors of the
measured radiances $\mathbf{y}_i$ ), which characterize the measurements, and the a priori CMs $\mathbf{S}_{ai}$ used in the retrievals, which
characterize the constraints. The expressions of $\mathbf{A}_i$ and $\mathbf{S}_i$ as a function of these two quantities are:

$$\mathbf{A}_i = \left( \mathbf{F}_i + \mathbf{S}_{ai}^{-1} \right)^{-1} \mathbf{F}_i \tag{1}$$

$$\mathbf{S}_i = \left( \mathbf{F}_i + \mathbf{S}_{ai}^{-1} \right)^{-1}. \tag{2}$$

We also recall that the $\mathbf{S}_i$ are the sum of two contributions: $\mathbf{S}_{ni}$, the CMs of the noise errors, and $\mathbf{S}_{si}$, the CMs of the smoothing errors, that are respectively equal to:

$$\mathbf{S}_{ni} = \left( \mathbf{F}_i + \mathbf{S}_{ai}^{-1} \right)^{-1} \mathbf{F}_i \left( \mathbf{F}_i + \mathbf{S}_{ai}^{-1} \right)^{-1} \tag{3}$$

$$\mathbf{S}_{si} = \left( \mathbf{F}_i + \mathbf{S}_{ai}^{-1} \right)^{-1} \mathbf{S}_{ai}^{-1} \left( \mathbf{F}_i + \mathbf{S}_{ai}^{-1} \right)^{-1} \tag{4}$$

and, as we can see from Eq. (2), the inverse matrices of $\mathbf{S}_i$ always exist.

Using the Kalman filter (Kalman, 1960, Rodgers, 2000) the new formula for the CDF was obtained in Ceccherini (2022) in the case of the fusion of two profiles. With an iterative procedure that adds one by one the extra profiles to the fused product (see Appendix A), it can be generalized to the fusion of $N$ retrieved profiles $\hat{\mathbf{x}}_i$ and expressed by the following formula:

$$\mathbf{x}_f = \left( \sum_{i=1}^{N} \mathbf{S}_i^{-1} \mathbf{A}_i + \mathbf{S}_a^{-1} \right)^{-1} \left( \sum_{i=1}^{N} \mathbf{S}_i^{-1} \boldsymbol{\alpha}_i + \mathbf{S}_a^{-1} \mathbf{x}_a \right), \tag{5}$$

where $\mathbf{x}_f$ is the fused profile, $\mathbf{x}_a$ and $\mathbf{S}_a$ are the a priori profile and its CM used to constrain the fused profile and

$$\boldsymbol{\alpha}_i = \hat{\mathbf{x}}_i - \mathbf{x}_{ai} + \mathbf{A}_i \mathbf{x}_{ai}, \tag{6}$$

$\mathbf{x}_{ai}$ being the a priori profiles used in the retrievals of the individual $\hat{\mathbf{x}}_i$, in general different among them and from $\mathbf{x}_a$. In the following we refer to the CDF formula given in Eq. (5) as CDF(2022).

From Eqs. (1-3) we see that we can express $\mathbf{S}_{ni}$ in terms of $\mathbf{A}_i$ and $\mathbf{S}_i$

$$\mathbf{S}_{ni} = \mathbf{S}_i \mathbf{A}_i^T = \mathbf{A}_i \mathbf{S}_i \tag{7}$$

and in the hypothesis that the CMs of the noise errors are nonsingular matrices we can obtain $\mathbf{S}_i^{-1}$

$$\mathbf{S}_i^{-1} = \mathbf{A}_i^T \mathbf{S}_{ni}^{-1}. \tag{8}$$

Substituting them in Eq. (5) we obtain the original formula for the CDF given in Ceccherini et al. (2015).

$$\mathbf{x}_f = \left( \sum_{i=1}^{N} \mathbf{A}_i^T \mathbf{S}_{ni}^{-1} \mathbf{A}_i + \mathbf{S}_a^{-1} \right)^{-1} \left( \sum_{i=1}^{N} \mathbf{A}_i^T \mathbf{S}_{ni}^{-1} \boldsymbol{\alpha}_i + \mathbf{S}_a^{-1} \mathbf{x}_a \right), \tag{9}$$

which, differently from Eq. (5), holds only in the case that the CMs of the noise errors $\mathbf{S}_{ni}$ are nonsingular matrices. Therefore, Eq. (5) is more general than Eq. (9). In the following we refer to the CDF formula given in Eq. (9) as CDF(2015). As already stated, the output of the CDF is not only the fused profile, but also its AKM and CM. The AKM and the CM of the fused profile calculated using Eq. (9) also contained the inverse of $\mathbf{S}_{ni}$ in the formulas (Ceccherini et al., 2015). We can now calculate these quantities for the products of Eq. (5) aiming at obtaining expressions that do not contain the inverse of matrices that may be singular. From Eq. (5) the AKM of $\mathbf{x}_f$ is given by:

$$\mathbf{A}_f = \frac{\partial \mathbf{x}_f}{\partial \mathbf{x}_t} = \left( \sum_{i=1}^{N} \mathbf{S}_i^{-1} \mathbf{A}_i + \mathbf{S}_a^{-1} \right)^{-1} \sum_{i=1}^{N} \mathbf{S}_i^{-1} \frac{\partial \boldsymbol{\alpha}_i}{\partial \mathbf{x}_t} = \left( \sum_{i=1}^{N} \mathbf{S}_i^{-1} \mathbf{A}_i + \mathbf{S}_a^{-1} \right)^{-1} \sum_{i=1}^{N} \mathbf{S}_i^{-1} \mathbf{A}_i, \tag{10}$$

where we have used Eq. (6) for the calculation of the derivatives.

The noise CM of $\mathbf{x}_f$ is obtained exploiting the fact that the noise CMs of $\boldsymbol{\alpha}_i$ are $\mathbf{S}_{ni}$, therefore,

$$\mathbf{S}_{nf} = \left( \sum_{i=1}^{N} \mathbf{S}_i^{-1} \mathbf{A}_i + \mathbf{S}_a^{-1} \right)^{-1} \sum_{i=1}^{N} \mathbf{S}_i^{-1} \mathbf{S}_{ni} \mathbf{S}_i^{-1} \left( \sum_{i=1}^{N} \mathbf{S}_i^{-1} \mathbf{A}_i + \mathbf{S}_a^{-1} \right)^{-1}. \tag{11}$$

Substituting $\mathbf{S}_{ni}$ given in Eq. (7) in Eq. (11), we obtain

$$\mathbf{S}_{\mathrm{nf}} = \left( \sum_{i=1}^{N} \mathbf{S}_i^{-1} \mathbf{A}_i + \mathbf{S}_{\mathrm{a}}^{-1} \right)^{-1} \sum_{i=1}^{N} \mathbf{S}_i^{-1} \mathbf{A}_i \left( \sum_{i=1}^{N} \mathbf{S}_i^{-1} \mathbf{A}_i + \mathbf{S}_{\mathrm{a}}^{-1} \right)^{-1}. \tag{12}$$

The CM of $\mathbf{x}_{\mathrm{f}}$ is obtained adding to Eq. (12) the CM of the smoothing errors

$$\mathbf{S}_{\mathrm{sf}} = \left( \sum_{i=1}^{N} \mathbf{S}_i^{-1} \mathbf{A}_i + \mathbf{S}_{\mathrm{a}}^{-1} \right)^{-1} \mathbf{S}_{\mathrm{a}}^{-1} \left( \sum_{i=1}^{N} \mathbf{S}_i^{-1} \mathbf{A}_i + \mathbf{S}_{\mathrm{a}}^{-1} \right)^{-1}, \tag{13}$$

obtaining

$$\mathbf{S}_{\mathrm{f}} = \mathbf{S}_{\mathrm{nf}} + \mathbf{S}_{\mathrm{sf}} = \left( \sum_{i=1}^{N} \mathbf{S}_i^{-1} \mathbf{A}_i + \mathbf{S}_{\mathrm{a}}^{-1} \right)^{-1}. \tag{14}$$

In this Section, following the recent results published in the literature, we started from the formula CDF(2022) and demonstrated that it is a generalization of CDF(2015). An alternative line of thought can also be followed. One can start from the formula CDF(2015), valid only in the hypothesis that the noise CMs are nonsingular matrices, and using Eq. (8) derive the formula CDF(2022). Noticing that the use of this formula does not require anymore the hypothesis that the CMs of the noise errors are nonsingular matrices, one can assume its general validity. The correctness of this assumption is then confirmed by the fact that CDF(2022) can also be obtained using the Kalman filter as shown in Ceccherini (2022).

In Appendix B we re-write some equations in a way that better highlights their physical meaning, although Eqs. (5) and (9) remain the CDF equations that can be used operationally.

## 2.2 Relationship between CDF(2022) and CDF(2015) with generalized inverse matrices

In the introduction we mentioned that, using the approximation of the generalized inverse matrices (Kalman, 1976), the original formula CDF(2015) can also be used in the case of $\mathbf{S}_{\mathrm{n}i}$ singular. Therefore, in this Section, we investigate the differences between CDF(2022) and CDF(2015) when in the latter the generalized inverse matrices of $\mathbf{S}_{\mathrm{n}i}$ are used. In Eq. (9) we replace the matrices $\mathbf{S}_{\mathrm{n}i}^{-1}$ with the generalized inverse matrices $\mathbf{S}_{\mathrm{n}i}^{\#}$

$$\mathbf{x}_{\mathrm{f}} = \left( \sum_{i=1}^{N} \mathbf{A}_i^{T} \mathbf{S}_{\mathrm{n}i}^{\#} \mathbf{A}_i + \mathbf{S}_{\mathrm{a}}^{-1} \right)^{-1} \left( \sum_{i=1}^{N} \mathbf{A}_i^{T} \mathbf{S}_{\mathrm{n}i}^{\#} \boldsymbol{\alpha}_i + \mathbf{S}_{\mathrm{a}}^{-1} \mathbf{x}_{\mathrm{a}} \right). \tag{15}$$

$\mathbf{S}_{\mathrm{n}i}^{\#}$ appear in two terms. For the first term it has already been demonstrated in the appendix of Ceccherini et al. (2012) that

$$\mathbf{A}_i^{T} \mathbf{S}_{\mathrm{n}i}^{\#} \mathbf{A}_i = \mathbf{F}_i = \mathbf{S}_i^{-1} \mathbf{A}_i, \tag{16}$$

where the second equality follows from Eqs. (1) and (2). Therefore, the first term is equal in the two CDF formulas.

We can elaborate the second term using Eqs. (1-3)

$$\mathbf{A}_i^{T} \mathbf{S}_{\mathrm{n}i}^{\#} = \mathbf{F}_i \left( \mathbf{F}_i + \mathbf{S}_{\mathrm{a}i}^{-1} \right)^{-1} \mathbf{S}_{\mathrm{n}i}^{\#} = \left( \mathbf{F}_i + \mathbf{S}_{\mathrm{a}i}^{-1} \right) \left( \mathbf{F}_i + \mathbf{S}_{\mathrm{a}i}^{-1} \right)^{-1} \mathbf{F}_i \left( \mathbf{F}_i + \mathbf{S}_{\mathrm{a}i}^{-1} \right)^{-1} \mathbf{S}_{\mathrm{n}i}^{\#} = \mathbf{S}_i^{-1} \mathbf{S}_{\mathrm{n}i} \mathbf{S}_{\mathrm{n}i}^{\#}, \tag{17}$$

which, in general, are different from $\mathbf{S}_i^{-1}$, because $\mathbf{S}_{\mathrm{n}i} \mathbf{S}_{\mathrm{n}i}^{\#}$ are different from the identity matrices when $\mathbf{S}_{\mathrm{n}i}$ are singular matrices.

Therefore, in the case of singular $\mathbf{S}_{\mathrm{n}i}$, the CDF(2015) used with the generalized inverse matrices of $\mathbf{S}_{\mathrm{n}i}$, Eq. (15), is equivalent to

$$\mathbf{x}_{\mathrm{f}} = \left( \sum_{i=1}^{N} \mathbf{S}_i^{-1} \mathbf{A}_i + \mathbf{S}_{\mathrm{a}}^{-1} \right)^{-1} \left( \sum_{i=1}^{N} \mathbf{S}_i^{-1} \mathbf{S}_{\mathrm{n}i} \mathbf{S}_{\mathrm{n}i}^{\#} \boldsymbol{\alpha}_i + \mathbf{S}_{\mathrm{a}}^{-1} \mathbf{x}_{\mathrm{a}} \right). \tag{18}$$

This equation shows that the CDF(2015) used with the generalized inverse matrices is an approximation of the more rigorous CDF(2022) and the quality of the approximation depends on how much $\mathbf{S}_{\mathrm{n}i} \mathbf{S}_{\mathrm{n}i}^{\#}$ is close to the identity matrix.

**2.3 The new formula in presence of coincidence and interpolation errors**

We know that in the applications of the CDF to real measurements it is often necessary to fuse vertical profiles measured on different grids and at either different times or locations, so that, interpolation and coincidence errors must also be considered. The expression of the CDF with interpolation and coincidence errors, that can be called the operational CDF, was calculated in Ceccherini et al. (2018) and was derived from the CDF(2015) that, as we have seen above, is not valid when there are singular matrices. In this Section, we show how the expression of the operational CDF can be written in a more general

form, using the CDF(2022) and exploiting the equivalence of CDF(2015) and CDF(2022) in the case that the CMs of the noise errors are nonsingular.

We start from the formula that deals with interpolation and coincidence errors, given in Ceccherini et al. (2018), based on the CDF(2015) and equal to:

$$\mathbf{x}_\mathrm{f} = \left( \sum_{i=1}^{N} \mathbf{R}_i^T \mathbf{A}_i^T \tilde{\mathbf{S}}_{\mathrm{n}i}^{-1} \mathbf{A}_i \mathbf{R}_i + \mathbf{S}_\mathrm{a}^{-1} \right)^{-1} \left( \sum_{i=1}^{N} \mathbf{R}_i^T \mathbf{A}_i^T \tilde{\mathbf{S}}_{\mathrm{n}i}^{-1} \tilde{\boldsymbol{\alpha}}_i + \mathbf{S}_\mathrm{a}^{-1} \mathbf{x}_\mathrm{a} \right), \tag{19}$$

where $\mathbf{R}_i$ are the generalized inverse matrices of the interpolation matrices $\mathbf{H}_i$, which interpolate the profiles from the

retrieval grids to the fusion grid. Furthermore,

$$\tilde{\boldsymbol{\alpha}}_i = \boldsymbol{\alpha}_i - \mathbf{A}_i \left( \mathbf{C}^{(i)} - \mathbf{R}_i \mathbf{C}^{(\mathrm{f})} \right) \mathbf{x}_{\mathrm{a,fine}} \tag{20}$$

$$\tilde{\mathbf{S}}_{\mathrm{n}i} = \mathbf{S}_{\mathrm{n}i} + \mathbf{A}_i \left( \mathbf{C}^{(i)} - \mathbf{R}_i \mathbf{C}^{(\mathrm{f})} \right) \mathbf{S}_{\mathrm{a,fine}} \left( \mathbf{C}^{(i)} - \mathbf{R}_i \mathbf{C}^{(\mathrm{f})} \right)^T \mathbf{A}_i^T + \mathbf{A}_i \mathbf{C}^{(i)} \mathbf{S}_{\mathrm{coin}} \mathbf{C}^{(i)T} \mathbf{A}_i^T, \tag{21}$$

where $\mathbf{x}_{\mathrm{a,fine}}$ is the a priori profile used to constrain the data fusion represented on a fine grid that includes all the levels of the fusion grid and of the $N$ retrievals grids. $\mathbf{C}^{(i)}$ and $\mathbf{C}^{(\mathrm{f})}$ are the sampling matrices from this fine grid to the grid of the $i$-th retrieval and to the fusion grid, respectively. $\mathbf{S}_{\mathrm{a,fine}}$ and $\mathbf{S}_{\mathrm{coin}}$ are respectively the fusion a priori CM and the CM describing the variability of the true profiles related to the measurements that we fuse: both CMs are represented on the fine grid. The

same limit of Eq. (9) applies also to Eq. (19) that, evidently, can be written only in the hypothesis that $\tilde{\mathbf{S}}_{\mathrm{n}i}$ are nonsingular matrices.

In order to write an equation similar to Eq. (7) for $\tilde{\mathbf{S}}_{\mathrm{n}i}$, we define the matrix $\tilde{\mathbf{S}}_i$

$$\tilde{\mathbf{S}}_i = \mathbf{S}_i + \mathbf{A}_i \left( \mathbf{C}^{(i)} - \mathbf{R}_i \mathbf{C}^{(\mathrm{f})} \right) \mathbf{S}_{\mathrm{a,fine}} \left( \mathbf{C}^{(i)} - \mathbf{R}_i \mathbf{C}^{(\mathrm{f})} \right)^T + \mathbf{A}_i \mathbf{C}^{(i)} \mathbf{S}_{\mathrm{coin}} \mathbf{C}^{(i)T} \tag{22}$$

and from Eqs. (7, 21 and 22) we see that the following equation holds

$$\tilde{\mathbf{S}}_{\mathrm{n}i} = \tilde{\mathbf{S}}_i \mathbf{A}_i^T . \tag{23}$$

We observe that the matrix $\tilde{\mathbf{S}}_i$ is not symmetric and, therefore, does not represent a CM. However, this only concerns the

physical meaning of the quantities and does not interfere with the validity of the equations. On the other hand, we can see from Eq. (21) that $\tilde{\mathbf{S}}_{\mathrm{n}i}$ is symmetric and, therefore, equal to its transpose, so that also the following equation holds:

$$\tilde{\mathbf{S}}_{\mathrm{n}i} = \mathbf{A}_i \tilde{\mathbf{S}}_i^T . \tag{24}$$

We substitute Eq. (23) in Eq. (19) and obtain:

$$\mathbf{x}_\mathrm{f} = \left( \sum_{i=1}^{N} \mathbf{R}_i^T \mathbf{A}_i^T \left( \tilde{\mathbf{S}}_i \mathbf{A}_i^T \right)^{-1} \mathbf{A}_i \mathbf{R}_i + \mathbf{S}_\mathrm{a}^{-1} \right)^{-1} \left( \sum_{i=1}^{N} \mathbf{R}_i^T \mathbf{A}_i^T \left( \tilde{\mathbf{S}}_i \mathbf{A}_i^T \right)^{-1} \tilde{\boldsymbol{\alpha}}_i + \mathbf{S}_\mathrm{a}^{-1} \mathbf{x}_\mathrm{a} \right). \tag{25}$$

From Eq. (23) we see that the hypothesis of $\tilde{\mathbf{S}}_{\mathrm{n}i}$ nonsingular implies that also $\mathbf{A}_i$ and $\tilde{\mathbf{S}}_i$ are nonsingular, therefore, from Eq. (25) we obtain the new formula for operational CDF that does no longer contain inverse of matrices that can be singular

$$\mathbf{x}_{\mathrm{f}} = \left( \sum_{i=1}^{N} \mathbf{R}_i^T \tilde{\mathbf{S}}_i^{-1} \mathbf{A}_i \mathbf{R}_i + \mathbf{S}_{\mathrm{a}}^{-1} \right)^{-1} \left( \sum_{i=1}^{N} \mathbf{R}_i^T \tilde{\mathbf{S}}_i^{-1} \tilde{\boldsymbol{\alpha}}_i + \mathbf{S}_{\mathrm{a}}^{-1} \mathbf{x}_{\mathrm{a}} \right). \tag{26}$$

It is simple to see that in case of absence of interpolation and coincidence errors (that is all the vertical grids coincide and $\mathbf{S}_{\mathrm{coin}}$ is zero) Eq. (26) becomes Eq. (5). Therefore, Eq. (26), which coincides with the operational CDF of Eq. (19) when $\tilde{\mathbf{S}}_{\mathrm{n}i}$ are nonsingular and coincides with the CDF(2022) in absence of interpolation and coincidence errors, can be used as the new operational CDF rigorously valid also when the noise CMs of the retrieved products are singular matrices.

We can also calculate the AKM and the CMs of the fused profile obtained using Eq. (26). The AKM of $\mathbf{x}_{\mathrm{f}}$ is given by

$$\mathbf{A}_{\mathrm{f}} = \frac{\partial \mathbf{x}_{\mathrm{f}}}{\partial \overline{\mathbf{x}}} = \left( \sum_{i=1}^{N} \mathbf{R}_i^T \tilde{\mathbf{S}}_i^{-1} \mathbf{A}_i \mathbf{R}_i + \mathbf{S}_{\mathrm{a}}^{-1} \right)^{-1} \sum_{i=1}^{N} \mathbf{R}_i^T \tilde{\mathbf{S}}_i^{-1} \frac{\partial \tilde{\boldsymbol{\alpha}}_i}{\partial \overline{\mathbf{x}}} = \left( \sum_{i=1}^{N} \mathbf{R}_i^T \tilde{\mathbf{S}}_i^{-1} \mathbf{A}_i \mathbf{R}_i + \mathbf{S}_{\mathrm{a}}^{-1} \right)^{-1} \sum_{i=1}^{N} \mathbf{R}_i^T \tilde{\mathbf{S}}_i^{-1} \mathbf{A}_i \mathbf{R}_i , \tag{27}$$

where $\overline{\mathbf{x}}$ is the unknown profile estimated by the data fusion, which for example can be the mean value of the true profiles of the measurements that are fused. The value of the derivative $\frac{\partial \tilde{\boldsymbol{\alpha}}_i}{\partial \overline{\mathbf{x}}} = \mathbf{A}_i \mathbf{R}_i$ is obtained from Eq. (17) of Ceccherini et al. (2018).

Exploiting the fact that the CMs of $\tilde{\boldsymbol{\alpha}}_i$ due to noise, interpolation and coincidence errors are $\tilde{\mathbf{S}}_{\mathrm{n}i}$ (Ceccherini et al., 2018), the corresponding CM of $\mathbf{x}_{\mathrm{f}}$ is equal to:

$$\mathbf{S}_{\mathrm{nf}} = \left( \sum_{i=1}^{N} \mathbf{R}_i^T \tilde{\mathbf{S}}_i^{-1} \mathbf{A}_i \mathbf{R}_i + \mathbf{S}_{\mathrm{a}}^{-1} \right)^{-1} \sum_{i=1}^{N} \mathbf{R}_i^T \tilde{\mathbf{S}}_i^{-1} \tilde{\mathbf{S}}_{\mathrm{n}i} \left( \tilde{\mathbf{S}}_i^{-1} \right)^T \mathbf{R}_i \left( \sum_{i=1}^{N} \mathbf{R}_i^T \mathbf{A}_i^T \left( \tilde{\mathbf{S}}_i^{-1} \right)^T \mathbf{R}_i + \mathbf{S}_{\mathrm{a}}^{-1} \right)^{-1} . \tag{28}$$

In order to simplify this equation, we consider the symmetric matrix given by the product $\tilde{\mathbf{S}}_i^{-1} \tilde{\mathbf{S}}_{\mathrm{n}i} \left( \tilde{\mathbf{S}}_i^{-1} \right)^T$ and use Eq. (24)

$$\tilde{\mathbf{S}}_i^{-1} \tilde{\mathbf{S}}_{\mathrm{n}i} \left( \tilde{\mathbf{S}}_i^{-1} \right)^T = \tilde{\mathbf{S}}_i^{-1} \mathbf{A}_i \tilde{\mathbf{S}}_i^T \left( \tilde{\mathbf{S}}_i^{-1} \right)^T = \tilde{\mathbf{S}}_i^{-1} \mathbf{A}_i = \mathbf{A}_i^T \left( \tilde{\mathbf{S}}_i^{-1} \right)^T , \tag{29}$$

where the last equality is obtained making the transpose and exploiting the fact that the matrix $\tilde{\mathbf{S}}_i^{-1} \tilde{\mathbf{S}}_{\mathrm{n}i} \left( \tilde{\mathbf{S}}_i^{-1} \right)^T$ is symmetric.

Using Eq. (29) in Eq. (28), the CM $\mathbf{S}_{\mathrm{nf}}$ becomes:

$$\mathbf{S}_{\mathrm{nf}} = \left( \sum_{i=1}^{N} \mathbf{R}_i^T \tilde{\mathbf{S}}_i^{-1} \mathbf{A}_i \mathbf{R}_i + \mathbf{S}_{\mathrm{a}}^{-1} \right)^{-1} \sum_{i=1}^{N} \mathbf{R}_i^T \tilde{\mathbf{S}}_i^{-1} \mathbf{A}_i \mathbf{R}_i \left( \sum_{i=1}^{N} \mathbf{R}_i^T \tilde{\mathbf{S}}_i^{-1} \mathbf{A}_i \mathbf{R}_i + \mathbf{S}_{\mathrm{a}}^{-1} \right)^{-1} . \tag{30}$$

The smoothing error CM of $\mathbf{x}_{\mathrm{f}}$ is equal to

$$\mathbf{S}_{\mathrm{sf}} = \left( \sum_{i=1}^{N} \mathbf{R}_i^T \tilde{\mathbf{S}}_i^{-1} \mathbf{A}_i \mathbf{R}_i + \mathbf{S}_{\mathrm{a}}^{-1} \right)^{-1} \mathbf{S}_{\mathrm{a}}^{-1} \left( \sum_{i=1}^{N} \mathbf{R}_i^T \tilde{\mathbf{S}}_i^{-1} \mathbf{A}_i \mathbf{R}_i + \mathbf{S}_{\mathrm{a}}^{-1} \right)^{-1} \tag{31}$$

and the CM of $\mathbf{x}_{\mathrm{f}}$ , obtained adding to the $\mathbf{S}_{\mathrm{nf}}$ given in Eq. (30) the smoothing error CM given in Eq. (31), is equal to:

$$\mathbf{S}_{\mathrm{f}} = \mathbf{S}_{\mathrm{nf}} + \mathbf{S}_{\mathrm{sf}} = \left( \sum_{i=1}^{N} \mathbf{R}_i^T \tilde{\mathbf{S}}_i^{-1} \mathbf{A}_i \mathbf{R}_i + \mathbf{S}_{\mathrm{a}}^{-1} \right)^{-1} . \tag{32}$$

This is an useful new equation that was not considered in Ceccherini et al. (2018).

**3 Performance comparison of the original and the new formula using an IASI measurement**

In this Section, we show an example of the error that we make using CDF(2015) instead of CDF(2022) on real data, using a METOP-B IASI ozone measurement acquired in the geolocation 43.45° of latitude and 10.77° of longitude, at 8:45:56 UTC of 18 October 2021.

In Fig. 1 we report the retrieved ozone profile with its a priori profile, errors and averaging kernels obtained with the Fast Optimal Retrieval on Layers for IASI (FORLI), described in Hurtmans et al. (2012) and Astoreca et al. (2014). This product was downloaded from the webpage "IASI Combined Sounding Products – Metop". FORLI retrieves the ozone profiles by means of the optimal estimation method and the radiative transfer calculation is performed using tabulated absorption cross sections at various pressures and temperatures in order to speed up the calculation time. The derivatives of the direct model

with respect to the state vector are computed analytically. The retrieval spectral range is 1025-1075 cm$^{-1}$ and the a priori information relies on the McPeters/Labow/Logan climatology of ozone profiles (McPeters et al., 2007). The ozone product of FORLI is a profile retrieved on 40 layers between surface and 40 km, with an extra layer from 40 km to the top of the atmosphere.

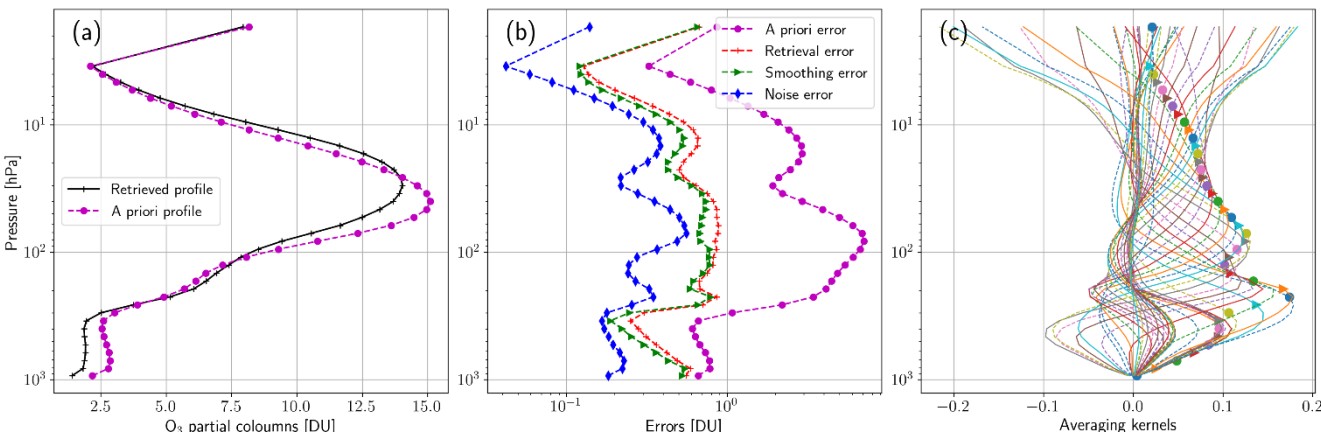

**Figure 1: Panel (a) shows the retrieved ozone profile and the a priori profile, panel (b) shows the errors and panel (c) shows the averaging kernels of the IASI measurement. The dots in panel (c) represent the diagonal values of the AKM.**

From Fig. 1 we can see that the profile used in this study is a typical product obtained with the optimal estimation method where most of the information is provided by the a priori as it results from the number of degrees of freedom, obtained by the sum of the diagonal values of the AKM, equal to 3.3 that is much smaller than the number of retrieved points.

In Fig. 2, we report the eigenvalues of $\mathbf{S}_i$ and $\mathbf{S}_{ni}$ for this IASI measurement calculated with the linalg.eigvals function of numpy Python 3 module version 1.20.2 (Numpy.linalg.eigvals).

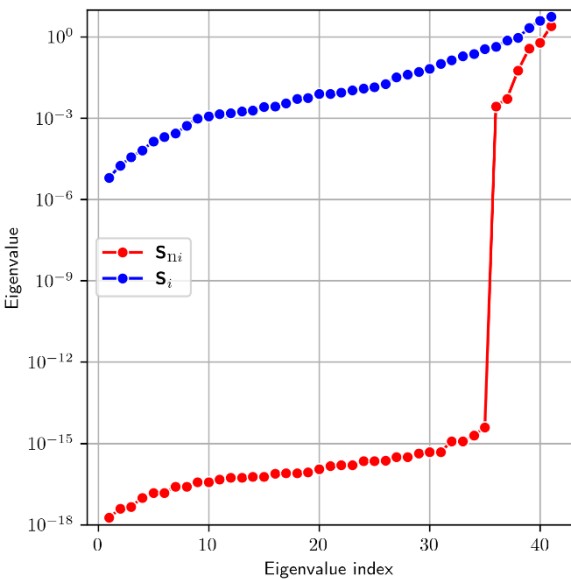

**Figure 2: Eigenvalues of the CMs $\mathbf{S}_i$ and $\mathbf{S}_{ni}$ of the IASI measurement.**

As expected, the eigenvalues of $\mathbf{S}_i$ are all different from zero, on the other hand, only 6 eigenvalues of $\mathbf{S}_{ni}$ have large values, while the others have values smaller than the numeric noise. The distribution of the eigenvalues of $\mathbf{S}_{ni}$ is due to the fact that the AKM and the retrieval error CM provided to the users are compressed (Astoreca et al., 2017) and are reconstructed using the 6 largest eigenvalues of the Fisher information matrix.

This product is used to perform a consistency check using the two CDF formulas, as described below.

The CDF formula can also be used to estimate, in the linear approximation, how the retrieved profile $\hat{\mathbf{x}}_i$ changes when the a priori profile $\mathbf{x}_{ai}$ and its CM $\mathbf{S}_{ai}$ are changed. This operation, explained in detail in Ceccherini et al. (2014), consists in using the CDF formula with a single input retrieved profile $\hat{\mathbf{x}}_i$, obtained with its a priori profile $\mathbf{x}_{ai}$ and a priori CM $\mathbf{S}_{ai}$, and with the application of a new constraint $\mathbf{x}'_{ai}$ and $\mathbf{S}'_{ai}$. The new profile $\hat{\mathbf{x}}_i{}'$, that is the original measurement with a new constraint, can be obtained using either CDF(2022) or CDF(2015):

$$\mathbf{x}'_{i\,\text{CDF(2022)}} = \left( \mathbf{S}_i^{-1}\mathbf{A}_i + \mathbf{S}'^{-1}_{ai} \right)^{-1} \left( \mathbf{S}_i^{-1}\boldsymbol{\alpha}_i + \mathbf{S}'^{-1}_{ai}\mathbf{x}'_{ai} \right) \tag{33}$$

$$\mathbf{x}'_{i\,\text{CDF(2015)}} = \left( \mathbf{A}_i^T\mathbf{S}_{ni}^{\#}\mathbf{A}_i + \mathbf{S}'^{-1}_{ai} \right)^{-1} \left( \mathbf{A}_i^T\mathbf{S}_{ni}^{\#}\boldsymbol{\alpha}_i + \mathbf{S}'^{-1}_{ai}\mathbf{x}'_{ai} \right), \tag{34}$$

where in the expression derived from CDF(2015) we have used the generalized inverse matrices of $\mathbf{S}_{ni}$ to deal with the most general case in which $\mathbf{S}_{ni}$ is singular.

When in Eqs. (33) and (34) we use a new constrain that is equal to the original one: $\mathbf{x}'_{ai} = \mathbf{x}_{ai}$ and $\mathbf{S}'_{ai} = \mathbf{S}_{ai}$, the formulas should provide the retrieved profile $\hat{\mathbf{x}}_i$. This is a check that we use to validate the self-consistency of the input data and that we can here use to assess the differences between the two CDF formulas.

Substituting $\boldsymbol{\alpha}_i$ from Eq. (6) in Eq. (33) and using Eqs. (2) and (16), we obtain that actually

$$\mathbf{x}'_{i\,\text{CDF(2022)}}\left[ \mathbf{x}'_{ai} = \mathbf{x}_{ai}, \quad \mathbf{S}'_{ai} = \mathbf{S}_{ai} \right] = \hat{\mathbf{x}}_i. \tag{35}$$

On the other hand, substituting $\boldsymbol{\alpha}_i$ from Eq. (6) in Eq. (34) we obtain:

$$\mathbf{x}'_{i\,\text{CDF(2015)}}\left[ \mathbf{x}'_{ai} = \mathbf{x}_{ai}, \quad \mathbf{S}'_{ai} = \mathbf{S}_{ai} \right] = \hat{\mathbf{x}}_i + \left[ \left( \mathbf{A}_i^T\mathbf{S}_{ni}^{\#}\mathbf{A}_i + \mathbf{S}_{ai}^{-1} \right)^{-1} \mathbf{A}_i^T\mathbf{S}_{ni}^{\#} - \mathbf{I} \right]\left( \hat{\mathbf{x}}_i - \mathbf{x}_{ai} \right), \tag{36}$$

where $\mathbf{I}$ is the identity matrix. The second term of Eq. (36) measures the error made using the generalized inverse and, using Eqs.(1, 3) and (16), we see that, in the case that $\mathbf{S}_{ni}$ is nonsingular, is equal to zero.

We have calculated the difference $\mathbf{x}'_{i\,\text{CDF(2015)}}\left[ \mathbf{x}'_{ai} = \mathbf{x}_{ai}, \quad \mathbf{S}'_{ai} = \mathbf{S}_{ai} \right] - \hat{\mathbf{x}}_i$ for several values of the threshold used to determine the eigenvalues that are neglected in the calculation of the generalized inverse matrix of $\mathbf{S}_{ni}$. In Fig. 3 we report the consistency test provided by this difference in the case of three values of the threshold that correspond to selecting, respectively, the 5, 6 and 7 largest eigenvalues. The generalized inverse matrices are calculated with the linalg.pinv function of numpy Python 3 module version 1.20.2 (Numpy.linalg.pinv), which calculates the Moore-Penrose pseudo-inverse of a matrix using the singular value decomposition and a threshold for the eigenvalues.

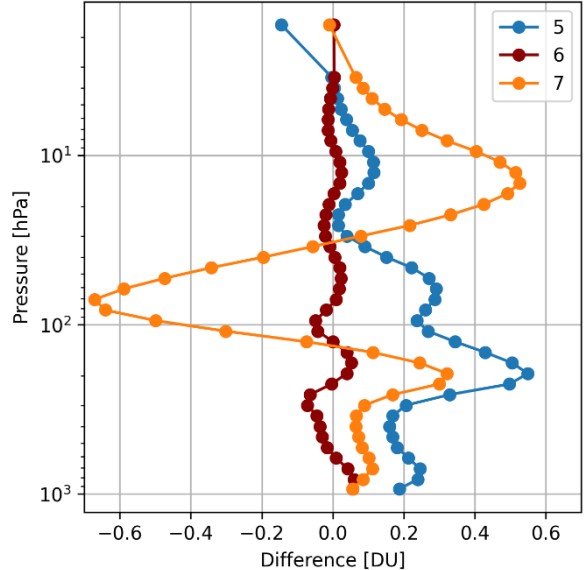

**Figure 3: Results of the consistency test with CDF(2015) considering only the 5, 6 and 7 largest eigenvalues in the calculation of the generalized inverse matrix of $\mathbf{S}_{\text{n}i}$.**

We can see that the smallest differences are obtained for the case of 6 eigenvalues, as expected from the distribution of the eigenvalues. The case of 5 eigenvalues is affected by the loss of useful information, on the other hand the case of 7 eigenvalues is affected by the amplification of the numeric noise. In this case, the choice of the threshold value can be simply done looking at Fig. 2, where the abrupt variation of the eigenvalues clearly indicates the threshold. In a general case, in which the variation of the eigenvalues is smooth, this test can be used to define the threshold for the eigenvalues choosing the value that minimizes the difference $\mathbf{x}'_{i\,\text{CDF(2015)}}\left[\mathbf{x}'_{\text{a}i}=\mathbf{x}_{\text{a}i},\ \mathbf{S}'_{\text{a}i}=\mathbf{S}_{\text{a}i}\right]-\hat{\mathbf{x}}_i$.

Using the optimum number of 6 eigenvalues for CDF(2015), in Fig. 4 we compare the differences $\mathbf{x}'_{i\,\text{CDF(2022)}}\left[\mathbf{x}'_{\text{a}i}=\mathbf{x}_{\text{a}i},\ \mathbf{S}'_{\text{a}i}=\mathbf{S}_{\text{a}i}\right]-\hat{\mathbf{x}}_i$ and $\mathbf{x}'_{i\,\text{CDF(2015)}}\left[\mathbf{x}'_{\text{a}i}=\mathbf{x}_{\text{a}i},\ \mathbf{S}'_{\text{a}i}=\mathbf{S}_{\text{a}i}\right]-\hat{\mathbf{x}}_i$ of the consistency test for the two CDF formulas with the retrieval error of the profile estimated by the square root of the diagonal elements of the CM $\mathbf{S}_i$.

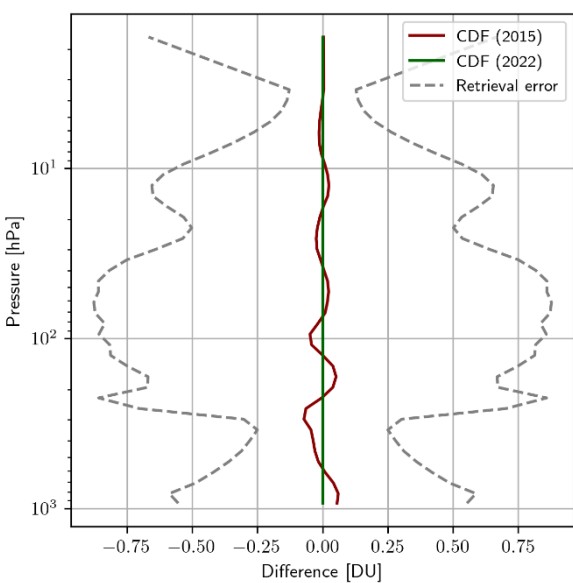

**Figure 4: Results of the consistency test applied to the IASI measurement for the two formulas CDF(2015) and CDF(2022) compared with the retrieval error of the profile.**

As expected the consistency test provides zero differences using CDF(2022) and detectable differences, although much smaller than the retrieval errors, are present when using CDF(2015). These differences are an estimate of the errors

introduced by CDF(2015) in the fusion process with respect to the results of CDF(2022). The comparison between Fig. 3 and Fig. 4 shows that the use of a number of eigenvalues that differs from the optimum value by one unity produces an error comparable with the retrieval error, therefore, it is very important to identify the optimum number of eigenvalues with the test described above.

The errors introduced by CDF(2015) depend on the compression used to represent the matrices in the files provided to the users. If less compression was applied to the data a greater number of eigenvalues could be considered in the calculation of the generalized inverse matrix of $\mathbf{S}_{ni}$ and the errors introduced by CDF(2015) would be further reduced.

When no compression is applied, the errors introduced by CDF(2015) are due to the numerical precision with which the data are provided, because the eigenvalues smaller than the numerical precision of the largest eigenvalue will usually only

contribute to the noise of the generalized inverse. Therefore, less compression and improved numerical precision can reduce the approximation introduced by CDF(2015).

## 4 Conclusions

The original CDF(2015) formula requires the calculation of the inverse matrices of the noise CMs $\mathbf{S}_{ni}$ of the input profiles and, therefore, can be rigorously applied only when these CMs are nonsingular. In the other cases, the CDF(2015) can still

be used replacing the inverse matrices of the noise CMs with the generalized inverse matrices, but the result is an approximation. Furthermore, a variable exists in this operation and a threshold has to be identified for the choice of how many eigenvalues are used in the calculation of the generalized inverse matrices.

A new formula CDF(2022) has been presented that contains the inverse matrices of the retrieval error CMs (the CMs that include both the noise and the smoothing errors), instead of the inverse matrices of the noise CMs. Since the retrieval error

CMs are always nonsingular matrices, the new formula can be used without resorting to generalized inverse matrices.

We deduced the analytical relationship between the two formulas and observed that the quality of the approximation provided by the old formula depends on how much $\mathbf{S}_{ni}\mathbf{S}_{ni}^{\#}$ is close to the identity matrix.

Furthermore, we have obtained the expression of the operational CDF(2022), which can handle interpolation and coincidence errors. The operational CDF(2022) is indispensable for the application of the CDF to real measurements, which

are often measured on different vertical grids and at different times and/or locations.

Finally, we have introduced a consistency check that can be used to define the threshold for the eigenvalues of the noise CMs and applied it to a real IASI measurement to evaluate the errors made using CDF(2015) instead of CDF(2022). We observed that in practice the errors introduced by the use of CDF(2015) are much smaller than the retrieval errors and depend on the data compression and numerical precision with which the data are provided to the users.

The errors made with the old CDF(2015) do not appear to be too large, even in the case of a significant data compression, however, the use of the new CDF(2022) and operational CDF(2022) is recommended for data fusion processing.

## Appendix A

In this appendix, we proof Eq. (5) that is the generalization to *N* profiles of the new formula for the CDF obtained using the Kalman filter in Ceccherini (2022) in the case of the fusion of two profiles.

At the basis of this proof there is the consideration that the product of the CDF is characterized by the same quantities that characterize the retrieval product: CMs, AKM and a priori information, therefore, it can be used as input for successive fusion operations.

Here we demonstrate that if Eq. (5) is valid for *N* it is valid also for *N*+1 and, since we know that is valid for *N*=2, using the induction principle, we deduce that is valid for any *N*.

We suppose to have fused $N$ profiles and, therefore, for hypothesis we have obtained the profile $\mathbf{x}_\mathrm{f}$ given by Eq. (5). Now we fuse $\mathbf{x}_\mathrm{f}$ with another profile $\hat{\mathbf{x}}_{N+1}$ using the Kalman filter. From Eq. (16) of Ceccherini (2022), we obtain the new fuse profile given by:

$$\mathbf{x}_\mathrm{f}^{(N+1)} = \left( \mathbf{S}_\mathrm{f}^{-1}\mathbf{A}_\mathrm{f} + \mathbf{S}_{N+1}^{-1}\mathbf{A}_{N+1} + \mathbf{S}_\mathrm{a}^{-1} \right)^{-1} \left( \mathbf{S}_\mathrm{f}^{-1}\boldsymbol{\alpha}_\mathrm{f} + \mathbf{S}_{N+1}^{-1}\boldsymbol{\alpha}_{N+1} + \mathbf{S}_\mathrm{a}^{-1}\mathbf{x}_\mathrm{a} \right), \tag{A1}$$

where $\boldsymbol{\alpha}_\mathrm{f}$ is given by:

$$\boldsymbol{\alpha}_\mathrm{f} = \mathbf{x}_\mathrm{f} - \mathbf{x}_\mathrm{a} + \mathbf{A}_\mathrm{f}\mathbf{x}_\mathrm{a} . \tag{A2}$$

Using Eqs. (10) and (14) we derive that:

$$\mathbf{S}_\mathrm{f}^{-1}\mathbf{A}_\mathrm{f} = \sum_{i=1}^{N} \mathbf{S}_i^{-1}\mathbf{A}_i . \tag{A3}$$

and using Eqs. (5, 10, 14) and (A2) we derive that:

$$\mathbf{S}_\mathrm{f}^{-1}\boldsymbol{\alpha}_\mathrm{f} = \sum_{i=1}^{N} \mathbf{S}_i^{-1}\boldsymbol{\alpha}_i . \tag{A4}$$

Substituting Eqs. (A3-A4) in Eq. (A1) we obtain:

$$\mathbf{x}_\mathrm{f}^{(N+1)} = \left( \sum_{i=1}^{N+1} \mathbf{S}_i^{-1}\mathbf{A}_i + \mathbf{S}_\mathrm{a}^{-1} \right)^{-1} \left( \sum_{i=1}^{N+1} \mathbf{S}_i^{-1}\boldsymbol{\alpha}_i + \mathbf{S}_\mathrm{a}^{-1}\mathbf{x}_\mathrm{a} \right), \tag{A5}$$

which is Eq. (5) written for the fusion of $N+1$ profiles. Therefore, as anticipated above, using the induction principle, we can state that Eq. (5) is valid for any $N$.

**Appendix B**

In this appendix, we re-write some equations of the CDF presented in the paper in a way that better highlights their physical meaning.

If we expand to the first order the relationships between the retrieved profiles $\hat{\mathbf{x}}_i$ and the true profile $\mathbf{x}_\mathrm{t}$ around the a priori profiles $\mathbf{x}_{\mathrm{a}i}$, we obtain:

$$\hat{\mathbf{x}}_i = \mathbf{x}_{\mathrm{a}i} + \mathbf{A}_i \left( \mathbf{x}_\mathrm{t} - \mathbf{x}_{\mathrm{a}i} \right) + \mathbf{G}_i\boldsymbol{\varepsilon}_i = \mathbf{A}_i\mathbf{x}_\mathrm{t} + \left( \mathbf{I} - \mathbf{A}_i \right)\mathbf{x}_{\mathrm{a}i} + \mathbf{G}_i\boldsymbol{\varepsilon}_i , \tag{B1}$$

where $\mathbf{G}_i = \left( \mathbf{K}_i^T\mathbf{S}_{\mathrm{ny}_i}^{-1}\mathbf{K}_i + \mathbf{S}_{\mathrm{a}i}^{-1} \right)^{-1}\mathbf{K}_i^T\mathbf{S}_{\mathrm{ny}_i}^{-1}$ are the gain matrices and $\boldsymbol{\varepsilon}_i$ are the noise errors of the measured radiances $\mathbf{y}_i$.

Using Eqs. (6) and (B1) we can re-write $\boldsymbol{\alpha}_i$ as:

$$\boldsymbol{\alpha}_i = \mathbf{A}_i\mathbf{x}_\mathrm{t} + \mathbf{G}_i\boldsymbol{\varepsilon}_i , \tag{B2}$$

that is $\boldsymbol{\alpha}_i$ is the true profile smoothed by the averaging kernels of the $i$-th measurement plus the error. Therefore, $\boldsymbol{\alpha}_i$ can be interpreted as a measurement of the true profile performed with the weighting functions given by the rows of $\mathbf{A}_i$.

Substituting Eq. (B2) in Eq. (5) we obtain:

$$\mathbf{x}_\mathrm{f} = \left( \sum_{i=1}^{N} \mathbf{S}_i^{-1}\mathbf{A}_i + \mathbf{S}_\mathrm{a}^{-1} \right)^{-1} \left( \sum_{i=1}^{N} \mathbf{S}_i^{-1}\mathbf{A}_i\mathbf{x}_\mathrm{t} + \mathbf{S}_\mathrm{a}^{-1}\mathbf{x}_\mathrm{a} + \sum_{i=1}^{N} \mathbf{K}_i^T\mathbf{S}_{\mathrm{ny}_i}^{-1}\boldsymbol{\varepsilon}_i \right) \tag{B3}$$

and using Eq. (16), Eq. (B3) becomes:

$$\mathbf{x}_\mathrm{f} = \left( \sum_{i=1}^{N} \mathbf{F}_i + \mathbf{S}_\mathrm{a}^{-1} \right)^{-1} \left( \sum_{i=1}^{N} \mathbf{F}_i\mathbf{x}_\mathrm{t} + \mathbf{S}_\mathrm{a}^{-1}\mathbf{x}_\mathrm{a} + \sum_{i=1}^{N} \mathbf{K}_i^T\mathbf{S}_{\mathrm{ny}_i}^{-1}\boldsymbol{\varepsilon}_i \right). \tag{B4}$$

Eq. (B4) clearly shows that the CDF profile is the weighted mean of the true profile, weighted $N$ times with the Fisher information matrices of the $N$ different measurements, and of the a priori profile weighted with the matrix $\mathbf{S}_a^{-1}$.

Using this formalism we can re-write also Eqs. (10) and (14), which give the expressions of the AKM and CM of the fused profile:

$$\mathbf{A}_f = \left( \sum_{i=1}^{N} \mathbf{F}_i + \mathbf{S}_a^{-1} \right)^{-1} \sum_{i=1}^{N} \mathbf{F}_i \qquad (B5)$$

$$\mathbf{S}_f = \left( \sum_{i=1}^{N} \mathbf{F}_i + \mathbf{S}_a^{-1} \right)^{-1} . \qquad (B6)$$

Eq. (B4) is equivalent to Eq. (5) and reveals the physical meaning of the CDF as a weighted mean of a set of measurements.
However, while Eq. (5) is expressed using the retrieval products ( $\boldsymbol{\alpha}_i$ quantities obtained from the retrieved profiles, AKMs and CMs) and, therefore, can be operatively used, the same does not apply to Eq. (B4) which is expressed using unknown quantities (such as the true profile and the errors).

As a final consideration, we notice that the CDF can be traced back to the general approach outlined in Section 4.1.1 of Rodgers (2000), once that the new linearized independent measurements $\boldsymbol{\alpha}_i$ have been introduced. Indeed, if in Eq. (4.20) of
Rodgers (2000) we replace the measurements $\mathbf{y}_i$ with $\boldsymbol{\alpha}_i$, the Jacobians $\mathbf{K}_i$ with $\mathbf{A}_i$ and the CMs $\mathbf{S}_{\varepsilon_i}$ with $\mathbf{S}_{ni}$, we obtain the CDF formula in the formalism of Eq. (9), apart from the difference that in Eq. (9) the a priori is made explicit. Therefore, the CDF can be interpreted as an optimal estimate obtained by all the considered measurements linearized around the individual solutions. However, the general formalism exposed in Section 4.1.1 of Rodgers (2000) cannot be directly applied to the profiles retrieved with the optimal estimation method because affected by the bias of the a priori and the merit of the
CDF is the individuation of the $\boldsymbol{\alpha}_i$ quantities that overcome this limitation.

*Data availability.* The IASI data used in the paper are available at the webpage "IASI Combined Sounding Products – Metop": https://navigator.eumetsat.int/product/EO:EUM:DAT:METOP:IASSND02, last access 21 October 2022. The results of the analysis performed on these data are available from the authors upon request.

*Author contributions.* SC derived the new formula of the CDF and extended it to the case in which coincidence and interpolation errors are present. He wrote the draft version of the paper. NZ contributed to the method extension, implemented the formulas in a Python code and performed the test on the IASI measurement. BC contributed to the interpretation of the results and made heavy revisions giving a coherent structure to the paper. All the authors revised the
paper.

*Competing interests.* The authors declare that they have no conflict of interest.

*Acknowledgments.* The authors thank EUMETSAT to produce and distribute the IASI data used in the paper.

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
