# Peer review of "An improved formula for the Complete Data Fusion"

_Atmospheric Measurement Techniques, 2022_

## Author Comment (AC1)

We thank the reviewer for the useful comments. In the following, we answer the specific comments (included in "**boldface**" for clarity) and, whenever required, we describe the related changes implemented in the revised manuscript. Indicated page and line numbers refer to the original version of the paper published on AMTD.

**Anonymous Referee #1**

**Review**

**General**

**Ceccherini et al. in this work provide an improved formalism for the complete data fusion of OE satellite retrievals, which they have developed earlier. The derivation of this new formalism is rigorous, but could be better situated with respect to the CDF development in previous work. Moreover, the demonstration of the 'improvement' in Section 3 looks quite poor. I therefore suggest major revisions, in line with the comments below.**

**Specific comments**

**Abstract and throughout the text: It sounds misleading to state that the CDF "combines independent measurements of an atmospheric vertical profile" – thus a single one – given that its extent (in space and time) is unknown and if later the formalism is extended "to include interpolation and coincidence errors, which must be considered when the profiles to be fused are measured on different vertical grids and at either different times or locations." The latter additionally implies that the indication of a single profile does not hold, in agreement with lines 26-28.**
In the revised version of the paper, we changed "combines independent measurements of an atmospheric vertical profile" in "combines independent measurements of atmospheric vertical profiles" in the abstract. On the other hand, at line 15, where reference to "an atmospheric vertical profile" is still present, it is specified that the given definition refers to how the CDF was introduced in Ceccherini et al. (2015).

**Eqs. (5), (10), and (12), are straightforwardly obtained from insertion of Eq. (8) of this work into Eqs. (5), (6), and (7) of Ceccherini et al., 2015, whereafter the need for nonsingular matrices (or the generalized inverses of singular matrices) can be dropped from the CDF formalism. This more intuitive line of thought should be discussed, with appropriate reference to what has already been achieved in Ceccherini, 2022.**
We followed an approach that was driven by the new information published in the literature and developed a rigorous demonstration; however, the line of thought suggested by the reviewer is probably more intuitive. Therefore, in the revised version of the paper we added a paragraph at the end of Section 2.1 where the alternative line of thought is presented.

**Given Sections 2.1 and 2.3, the conclusion of Section 2.2 is quite trivial and could be dropped.**
The last sentence of Section 2.2 wants to provide a comment to the previous Eq. (18) rather than a conclusion. In the revised version of the paper we modified this sentence to make this more evident.

**A nice plus with respect to previous work is the brief expression for the full CM of the CDF as expressed in Eq. (14) and its extension in the presence of coincidence and interpolation errors by Eq. (32). Following up on the previous comment, these findings could be better situated with respect to the equations in Ceccherini et al., 2018.**

A sentence was added in the revised paper at the end of Section 2.3 as a comment to Eq. (32).

**In order to be exhaustive, several things are missing from Section 3. It would at least require (1) a paragraph on the IASI data retrieval (lines 150-151), (2) information on the definition of the IASI profiles (cf. Figure 1, partial columns for which layers?), (3) a view on the AKM and CM shapes for the IASI retrieval and a discussion of how representative these are for OE retrievals, (4) expressions for how eigenvalues and generalized inverses are calculated.**

In the revised version of the paper, we added all the information requested by the reviewer. We modified also Fig. 1 in order to show the a priori profile, the AKM and the errors.

Reference to McPeters et al., 2007 was added in order to describe the a priori information used in the retrieval of the IASI measurement.

References to the Python functions used to calculate the eigenvalues and the generalized inverses were added.

**Given that (lines 159-161) "The distribution of the eigenvalues of S_ni is due to the fact that the AKM and the retrieval error CM provided to the users are compressed (Astoreca et al., 2017) and are reconstructed using the 6 largest eigenvalues of the Fisher information matrix." the result shown in Figure 3 does not come as a surprise. Figure 4 is then created for "the optimum number of eigenvalues" (please mention six explicitly). The most important conclusion of this work, however, is the following (last sentence of Section 4): "The use of the new CDF(2021) and operational CDF(2021) is recommended for data fusion processing, but the errors made with the old CDF(2015) do not appear to be important, even in the case of a significant data compression." Numbers should be given with this statement. In order to do so, Figure 4 should be reproduced for a range of generalized inverses (or numbers of eigenvalues), including the popular Moore-Penrose pseudo-inverse, to quantify what the effect of this choice is on the error that has been made by using CDF(2015). And finally, how does this error relate to the full CM S_f upon CDF of an increasing number of retrieved profiles? That should be the full merit of Section 3.**

In the revised version of the paper we explicitly mention six as the optimum number of eigenvalues, furthermore we specify that the calculation of the generalized inverse is made with the Moore-Penrose pseudo-inverse of a matrix using its singular value decomposition and a threshold for the eigenvalues (the Moore-Penrose pseudo inverse without a threshold does not have a physical meaning).

The effect of choosing a wrong threshold for the eigenvalues is shown in Fig. 3 and the comparison of this figure with Fig. 4 is sufficient to highlight that to be wrong in the number of eigenvalues of one unity produces an error comparable with the retrieval error. We added this consideration in the revised version of the paper at the end of the paragraph after Fig. 4. We prefer to make this comment and maintain Fig. 4 as simple as possible.

The question "how does this error relate to the full CM S_f upon CDF of an increasing number of retrieved profiles?" is quite important, however, it is difficult to provide examples that are valid for all the possible cases encountered with real data. The error of the fused profile decreases more slowly than the square root of the number of fusing profiles because: a) part of the information is used to increase the number of degrees of freedom and b) the contribution of the a priori errors to the components that are not measured does not decrease. On the other hand, the error due to the generalized inverse does in general decrease with the square root of the number of fusing profiles, but also in this case a lower decrease may be encountered if systematic effects are present. We considered making this comment to the paper, but we think that it would increase the complexity more than the information.

**Technical corrections**

**Lines 23-25: "The method is equivalent to the simultaneous retrieval of all the measurements that are combined when the linear approximation of the forward models is appropriate in the variability range of the results of the individual retrievals." A proof or reference is needed for this statement.**
The proof is given in the Appendix of Ceccherini et al. (2015). We added the reference in the revised version of the paper. Furthermore, we rephrased the sentence.

**References to (Ceccherini, 2021) should be updated to (Ceccherini, 2022).**
In the revised version of the paper, we updated the references and also substituted CDF(2021) with CDF(2022).

**Line 63: AKM and CM as a 'measure' for sensitivities and retrieval errors, respectively, does not sound correct. These are retrieval quantities that define / determine the sensitivities and errors.**
In the revised version of the paper, we replaced "measure" with "determine".

**Line 73: "With an iterative procedure that adds one by one the extra profiles to the fused product…" A proof or reference is needed for this statement.**
In the revised version of the paper, a proof of this statement is given in Appendix A.

**The IASI data providers are not co-authors nor acknowledged, which seems inappropriate.**
In the acknowledgments of the revised version of the paper, we thank EUMETSAT that is the originator and the distributor of the IASI data used in the paper.

---

## Author Comment (AC2)

We thank the reviewer for the useful comments. In the following, we answer the specific comments (included in "**boldface**" for clarity) and, whenever required, we describe the related changes implemented in the revised manuscript. Indicated page and line numbers refer to the original version of the paper published on AMTD.

**Anonymous Referee #2**

**Review**

**GENERAL COMMENTS**

**The paper is well written and describes the employed methods in sufficient detail. With the last section, the method is applied to a practical example and the differences are examined.**

**The topic fits the journal well.**

**The method, as extended in Sect. 2.3 is particularly useful when joining satellite measurements taken on different grids and with co-location errors, where only certain diagnostic matrices are provided.**
**I suggest publication after properly positioning the CDF method as multi-variate inverse CM-weighted mean and addressing the major and specific comments below.**
We have considered all the major comments listed below in order to show that, as suggested by the reviewer, the CDF can be described as a multivariate inverse CM weighted mean. Therefore, in the revised version of the paper, we have used the equations suggested by the reviewer, which provide useful information on the physical meaning of the CDF. However, these equations cannot be used to perform data fusion and we preferred to put this discussion in an appendix (Appendix B), in order to not interrupt the flux of the reasoning.

**MAJOR COMMENTS**
==============

**line 76**
* * *
**I think it might help here the understanding to introduce the relation of**
$\hat{x} = A\, x\_true + (I-A)\, x\_a + G\, \epsilon$
**as this shows more readily the nature of the formula: a weighted average of the true state transformed by the different measurement characteristics of the involved instruments:**
$x\_f = (\sum S^{-1}\_i A\_i + S\_a^{-1})^{-1} (\sum S^{-1}\_i (A\_i\, x\_true,i + G\_i\, \eps\_i) + S\_a^{-1}\, x\_a).$
**which also leads pretty naturally to the derivation of the aggregated averaging kernel matrix.**
In the revised version of the paper, we introduced these two equations in Appendix B.

**The formula above as well as (10) - (12) can also be simplified drastically by exploiting that**
$S\_i^{-1}A\_i = F\_i,$
**which mathematically is very reasonable and fits well to the general framework of optimal estimation and Kalman filtering.**
In the Appendix B we wrote the CDF formulas as suggested by the reviewer and discussed them.

**Is the whole method, in its given form, not fully identical to a "simple/straightforward" linear optimal estimation/maximum likelihood estimate of all involved instruments *linearized* around the individual solutions? Which is indeed very reasonable, but not really a "new" method.**
**The new mathematical description makes this pretty obvious in contrast to the original, more convoluted formula.**
We agree with the reviewer that the method can be seen as an optimal estimate obtained by all the considered measurements linearized around the individual solutions. We wrote this in the Appendix B.

**The given mathematical notation can be argued for due to the information supplied by typical retrieval products, but both forms, the "standard" form using the (inverse) Fisher information matrix as weight in a weighted mean and the given form should be described and compared against each other.**
In the Appendix B of the revised version of the paper we wrote the equations of CDF using the Fisher information matrix as weight in a weighted mean and compared these expressions with the original ones given in the manuscript.

**The authors should discuss this and how it differs (or not) from the method described, e.g., by Rodgers in Sect. 4.1.1.**
In The Appendix B of the revised version of the paper we discussed the connection between CDF and the method described in Section 4.1.1 of Rodgers (2000).

**SPECIFIC COMMENTS**
=================

**line 23**
* * *
**You stated that the method delivers the same result as a simultaneous retrieval, so in what respect or in relation to what can its quality be better?**
The implementation of the CDF is much simpler than that of the simultaneous retrieval. Indeed, the simultaneous retrieval requires to integrate into a single inversion system the radiative transfer models capable to simulate the measurements of the different sensors involved in the simultaneous inversion. Furthermore, the simultaneous retrieval requires the simultaneous access to all the (Level 1) measurements used in the inversion, thus implying the need to handle relevant data volumes. These characteristics complicate the implementation of the simultaneous retrieval and increase the necessary computational resources. The CDF overcomes these complications by combining the Level 2 products supplied by the individual retrieval processors of the measurements.
We added this consideration in the revised version of the paper.

**line 48**
* * *
**Didn't you just state that the formula was introduced by Ceccherini (2021)? So it isn't introduced here, "only" discussed in greater detail?**
In Ceccherini (2022) the formula was derived only for the fusion of two profiles, here we generalize the formula for any number of profiles and discuss it in greater detail.

In the revised version of the paper, we change the sentence specifying better what we have done here. Furthermore, we change also line 44, specifying that in Ceccherini (2022) the formula was derived only for the fusion of two profiles.

**In fact, Ceccherini (2021) seems to suggest that the formula was introduced by Schneider (2021)?**
The correct statement is that: starting from the Kalman filter method as applied in Schneider (2022), Ceccherini (2022) derived the new formula of the CDF for the fusion of two profiles.

**I think the historical development and relationship between the papers and methods should be discussed in slightly more detail than given here, taking into account in particular other peoples contributions.**
In the revised version of the paper, we changed the sentence and added the reference to Schneider et al. (2022).

**line 233**
* * *
**Is the Python code with a reference implementation available? I.e. can the results of Section 3 be reproduced?**
The Python code used in the paper is not user friendly and, accordingly, is not made available, but the results in Section 3 can be easily reproduced implementing formulas in Eq. (33-34). In the revised version of the paper, we provide the references to the Python functions used to calculate the eigenvalues and the generalized inverses.

**MINOR REMARKS**
=============

**line 7**
* * *
**Who has proposed it?**
It was proposed in Ceccherini (2022), but since in the abstract reference citations have to be avoided, the reader finds this information in the introduction, that in the revised version has been changed to better clarify the historical development and relationship between the papers and the method (see answer to the comment above).

**line 30**
* * *
**Performances ... have -> performance has**
In the revised version of the paper, we changed the sentence as suggested by the reviewer.

**line 39/43**
* * *
**I would say that while Rodgers provides a very useful discussion on the use of Kalman filters for the use case at hand, it is not a suitable reference without also giving (Kalman, 1960; see Rodgers). Are the references in lines 39 and 43 switched?**

In the revised version of the paper, we added the reference to Kalman (1960).
The reference in lines 39 and 43 are both correct.

**line 66**
**-------**

**The readability of the formulas could be greatly improved when the "^-1" notation of the involved matrices would be above the index, not after it.**
In the revised version of the paper, we moved the "^-1", "*T*" and "#" notations of the involved matrices from after the index to above it.

---

## Author Response (AR2)

As required by the preceding review file validation, we modified figure #1 in order to ensure that the used colour schemes allow readers with colour vision deficiencies to correctly interpret the findings.

Best regards

Simone Ceccherini